# QTL Analysis of Yield and End-Use Quality Traits in Texas Hard Red Winter Wheat

Mehmet Dogan [1,2,†], Zhen Wang [1,2], Mustafa Cerit [1,2,‡], Jorge L. Valenzuela-Antelo [1,2,§], Smit Dhakal [1,‖], Chenggen Chu [1,¶], Qingwu Xue [1], Amir M. H. Ibrahim [2], Jackie C. Rudd [1], Amy Bernardo [3], Paul St. Amand [3], Guihua Bai [3], Hongbin Zhang [2] and Shuyu Liu [1,*]

1   Texas A&M AgriLife Research and Extension Center, 6500 Amarillo Blvd. W., Amarillo, TX 79106, USA
2   Department of Soil and Crop Sciences, Texas A&M University, 370 Olsen Blvd, College Station, TX 77843, USA
3   USDA-ARS, Hard Winter Wheat Genetics Research Unit, Throckmorton Hall 4008, Manhattan, KS 66506, USA
*   Correspondence: shuyu.liu@ag.tamu.edu
†   Present address: Field Crops Central Research Institute, Sehit Cem Ersever Cd., Yenimahalle, Ankara 06170, Türkiye.
‡   Present address: Aegean Agricultural Research Institute, Canakkale Asfaltı Cd., Menemen, Izmir 35661, Türkiye.
§   Present address: Bayer Crop Science Mexico, Tlajomulco Research Station, Tlajomulco de Zuniga, Jalisco 45660, Mexico.
‖   Present address: PowerPollen, 27253 US HWY 69, Ames, IA 50010, USA.
¶   Present address: USDA-ARS, Sugarbeet, and Potato Research Unit, 1616 Albrecht Blvd N, Fargo, ND 58102, USA.

**Abstract:** Genetic dissection of complex traits by quantitative trait locus (QTL) analysis permits the understanding of the genotypic effects of QTL, interactions between QTLs, and QTL-by-environment interactions in wheat. This study aimed to identify the QTL linked to yield, its components, end-use quality traits including kernel, flour, and dough rheology, and related agronomic traits under dryland and irrigated conditions. A mapping population of 179 $F_{2:6}$ recombinant inbred lines (RILs) derived from 'TAM 111'/'TX05A001822' was evaluated for these traits to investigate their genetic stability and phenotypic plasticity using 2658 single nucleotide polymorphisms (SNPs) with 35 linkage groups. Traits associated with chromosome regions were detected for individual and across-environment QTL by inclusive composite interval mapping. A total of 30 QTL regions were identified, including 14 consistent QTLs mapped on 11 chromosomes and six pleiotropic QTLs mapped on 5 chromosomes. Three consistent QTLs in chromosomes 1A, 3B, and 6D might be novel. Three major QTLs with both consistent and pleiotropic effects were co-localized with known genes. The first QTL for dough mixing properties was physically clustered around *Glu-D1* and had an phenotypic variation explained (PVE) up to 31.3%. The second QTL for kernel-related traits was physically close to the *TaCWI-4A* (cell wall invertase) gene, which influences the thousand kernel weight, heading date, and harvest index, with a PVE of up to 12.3%. The third QTL, which was colocalized with the *TaCWI-5D* gene for kernel traits, was identified with a PVE of 6.7%. Epistasis was also detected, but major QTLs were not involved in significant epistasis or interactions with environmental effects. The current study provided new information that is useful for enhanced wheat breeding, which will benefit from the deployment of the favorable alleles for end-use quality, yield, and other agronomic traits in wheat-breeding programs through marker-assisted selection.

**Keywords:** bread wheat; quantitative trait loci; favorable allele; end-use quality; dough rheology; yield components

## 1. Introduction

Approximately 20% of proteins and calories in the global human food consumption are contributed by wheat products such as bread, cookies, pastries, pasta, and noodles [1,2]. The hard red winter wheat (HRWW) (*Triticum aestivum* L.) is one of the most popular

wheat classes in the US market. It is distinguished from other classes due to its unique combination of characteristics in terms of grain morphology and quality parameters, such as kernel hardness, protein content, color, etc. [3]. HRWW is the most-produced wheat in the USA and is known worldwide for its excellent end-use quality, including its high protein content, outstanding milling and baking properties, and great yield potential under dry-land conditions [4].

High-yielding wheat cultivars with desirable end-use qualities have long been major objectives of wheat breeding to meet the demands of consumers and the baking industry. The wheat grain market requires good morphological characteristics, such as uniform grain size and high-test weight, while food producers have more interest in processing-quality characteristics [5]. However, laborious, expensive, and time-consuming laboratory measurements of end-use quality parameters have impeded the early generation selection for the development of varieties with a high end-use quality. Therefore, molecular markers closely linked to end-use quality parameters are promising for facilitating their phenotypic evaluation for early generation breeding materials [6]. A QTL mapping analysis using molecular markers can dissect the genetic architecture of the traits from the complex environmental effects. Several methods have been developed for end-use quality measurements, such as the single kernel characterization system (SKCS) to measure kernel-related traits, near infrared reflectance (NIR) spectrometry to measure protein and ash contents, and mixograph to measure dough rheology parameters [7]. Kernel hardness determines the subclass of bread wheat on market based on the energy required to break grains, which is mainly regulated by two alleles at the Ha locus (*Pina* and *Pinb*) on chromosome 5DS [8,9]. Gluten components determine the dough extensibility and strength. Gluten is composited by storage protein glutenins and gliadins. These are mainly regulated by the loci that control high molecular weight glutenin subunits (HMWGs), *Glu-A1*, *Glu-B1*, and *Glu-D1*; low molecular weight glutenin subunits (LMWGs), *Glu-A3*, *Glu-B3*, and *Glu-D3*; and gliadins [10,11]. Mixograph is used to determine dough rheology, and farinograph and alveograph are used to measure gluten strength. Long mixograph midline peak time, high mixograph midline peak height, and wide mixograph midline tail width corresponding to a strong gluten strength are more desirable for bread making [12]. Additionally, several previous studies showed that the QTLs for mixograph dough properties were mainly co-localized with the glutenin loci (*Glu–A1*, *Glu-B1*, and *Glu-D1*) [7,12–24].

A high grain yield is an essential objective in wheat breeding. It is genetically complex and readily affected by multiple environmental factors. The improvement of the grain yield depends on yield components, including the thousand kernel weight (TKW), kernels per spike (KPS), the number of spikes per square meter (SPM), and related agronomical traits such as the plant height (PH) and heading date (HD) [25–28]. Several yield-related and morphological traits have displayed higher heritability than grain yield and have shown strong positive correlations with grain yield [28–30]. For instance, PH is regulated by number of dwarf genes (*Rth* genes) [31,32] that have pleiotropic effects on the grain yield by influencing the lodging and harvest index [33]. HD is regulated by vernalization (*Vrn*), photoperiod response (*Ppd*), and earliness per se (*Eps*) genes [27,34,35]. Other researches reported that a QTL linked to test weight (TW), HD, and grain yield on chromosome 7B was colocalized with the *Vrn-B3* gene, a QTL related to the plant height, test weight, and yield on chromosome 7D was colocalized with the *Vrn-D3* gene, and a QTL for grain yield identified on chromosome 1A was linked to the early flowering gene *Elf3* [30,36,37].

Identifying genomic regions or QTLs associated with quantitative traits through linkage mapping is an effective approach to dissecting the genetic mechanisms of complex traits and to developing molecular tools that assist in pyramiding the favorable alleles of genes in wheat breeding. Therefore, this study aims to identify the pleiotropic and consistent QTLs linked to kernel, end-use quality, yield-related, and agronomical traits in a biparental mapping population derived from 'TAM 111' and 'TX05A001822' under diverse environmental conditions.

## 2. Materials and Methods

### 2.1. Plant Materials and Field Trials

A recombinant inbred line (RIL) population consisting of 179 $F_{2:6}$ lines derived from the cross between 'TAM 111' and 'TX05A001822', which are both hard red winter wheat (HRWW), was developed by Texas A&M AgriLife Research. TAM 111 is well-known for its excellent yield potential, drought tolerance, and high adaptability [38–40]. TX05A001822 is an unreleased breeding line that is outstanding in several end-use quality traits [41]. For instance, TX05A001822 had a milling score of 4.6 and a baking score of 2.8 that ranked at the third and fourth position, respectively, among the 22 breeding lines tested in the 2010 regional wheat quality test. With lower values than TX05A001822, TAM 111 had a milling score of 4.2 and a baking score of 2.2. TX05A001822 scored at 3.9, 4.2, 14.7, and 4.9 for the quality traits mixing tolerance, baking quality, protein content, and bread loaf volume, respectively. These scores were better than those achieved by TAM 111, which received the corresponding scores of 3.2, 2.7, 13.5, and 3.2.

Regarding quality and yield evaluation, the RIL population and its parental lines were grown in seven environments based on year and location combinations. These included locations of Chillicothe, TX (34°07′ N, 99°18′ W) (designated as 17CH), and Bushland, TX (35°06′ N, 102°27′ W) (designated as 17BSP67 and 17BSP100), in the 2016–2017 growing season, and Dumas, TX (35°51′ N, 101°58′ W) (designated as 18DMS), McGregor, TX (31°27′ N, 97°23′ W) (designated as18MCG), Bushland, TX (designated as 18BSP100), and Uvalde, TX (29°22′ N, 99°83′ W) (designated as 18UVL), in the 2017–2018 growing season. The experiments at the locations 18DMS, 17BSP67, 17BSP100, and 18BSP100 were irrigated, while the other two years/locations, including 17CH, 18MCG, and 18UVL, were rain-fed. Based on the evapotranspiration (ET) demand, the irrigation levels were set at a 67% ET demand at 17BSP67 (Bushland, TX) and a 100% ET demand at 17BSP100 and 18BSP100 (Bushland, TX). The plot size for the irrigated field was 3.05 m long and 1.52 m wide, while the plot size in the drylands was 4.57 m by 1.52 m. The experiment in each location used an alpha-lattice design with two replications.

### 2.2. Phenotypic Data Collection

Data for agronomical traits, including the grain yield (YLD, $g/m^2$), were collected from all five environments (17CH, 18BSP100, 18DMS, 18MCG, and 18UVL). Data for the heading date (HD, days) and plant height (PH, cm) were obtained from the four environments (17BSP100, 18BSP100, 18DMS, and 18MCG) (Table S1). The YLD was measured by harvesting the whole plot with a combined harvester and was converted to $g/m^2$. HD was recorded, based on the Julian calendar, to count the number of days from January 1st to the date at which 50% of the plants in a plot were headed. PH was measured from the ground to the top of the spike without awn at the fully matured stage.

All samples from three environments (17CH, 18DMS, and 18MCG) were evaluated for end-use quality traits following the procedure by Dhakal et al. [23] and Yu et al. [24] at the Texas A&M University Quality Laboratory in College Station, TX. However, in terms of kernel characteristics, data from an additional environment (18BSP100) was included (Table S1). As tests are destructive, time-consuming, and expensive, all quality traits were evaluated using samples from only one replication of each environment to measure the quality parameters. The Perten Model Single Kernel Characterization System (SKCS) 4100 (Perten Instruments North America Inc., Springfield, IL, USA) was used to evaluate kernel hardness index (HARD, %), kernel diameter (DIAM, mm), single kernel weight (SKW, mg), and moisture index based on 300 individual kernels [42]. Flour was extracted from the milled grain of each sample and reported as flour yield (FYLD, %). Near-infrared reflectance (NIR—Perten Model NIRS DA7250 (Perten Instruments North America Inc., Springfield, IL, USA)) was used to measure the flour protein content (PROT, %) and ash content (ASH, %) at 14% moisture [43]. A mixograph (Mixograph National Manufacturing CO, Lincoln, NE, USA) was assigned to measure the dough mixing properties, including midline peak time (MLPT, min), midline peak value (MLPV, %), midline peak width (MLPW, %), midline

right slope (MLRS, % min$^{-1}$), midline tail width (MLTW,%), midline time X time (MLTXT, min), and midline time X width (MLTXW, %) [44].

Yield-related traits were evaluated as described by Assanga et al. [45] and Yang et al. [26]. Biomass samples from an inner half-meter row were taken in three environments (17BSP67, 18BSP100, and 18DMS) (Table S1), oven-dried at 60 °C for 72 h, and weighed to determine the total dry biomass (BM, g/m$^2$) and the grain yield of biomass samples (BMYLD, g/m$^2$). Furthermore, the harvest index (HI, %) was calculated by dividing the BMYLD by the total weight of sample BM; the spikes per square meter (SPM, spike m$^{-2}$) was calculated by counting the number of heads of the samples for BM; the thousand kernel weight (TKW, g) was measured by weighing 200 seeds per sample; the kernels per spike (KPS, kernels spike$^{-1}$) was calculated using the BMYLD, TKW, and SPM; the single-head dry weight (SHDW, mg head$^{-1}$) was calculated by dividing the total dry head weight by the number of heads; and the single-head grain weight (SHGW, mg head$^{-1}$) was calculated by dividing the total BMYLD by the number of heads. In addition, approximately 10 g of samples of the seeds from two environments (18BSP100 and 18DMS) (Table S1) were randomly taken and scanned using the scanner (HP 11956A, Hewlett-Packard, Palo Alto, CA, USA) to determine the kernel area (KAREA, mm$^{-2}$), kernel perimeter (KPERI, mm), kernel length (KLEN, mm), and kernel width (KWID, mm) using the software GrainScan, version 1.0 [46].

### 2.3. Statistical Analysis

Outliers in the phenotypic data were removed using the statistical software JMP Pro 16 [47]. Subsequently, the restricted maximum likelihood method (REML) was used to estimate the variance components and best linear unbiased predictors (BLUPs) for individual and multi-environmental data by assuming a fully random procedure in JMP software, v.16. The BLUP values of the traits were used to determine the significance of the genotype (G), environment (E), and genotype-by-environment interaction (GEI) variance components. Two different models were used to analyze the trait data. For all traits except end-use quality parameters, the following model was applied:

$$Y_{ijkl} = \mu + Gen_i + Env_j + Rep_k(Env_j) + Iblock_l(Rep_k xEnv_j) + Env_j xGen_i + \epsilon_{ijkl}$$

where $\mu$ is the trait mean, $Gen_i$ is the effect of the i$^{th}$ genotype, $Env_j$ is the effect of the j$^{th}$ environment, $Rep_k$ is the effect of the k$^{th}$ replication, $Iblock_l$ is the effect of the l$^{th}$ iblock, $Env_j xGen_i$ is the effect of the genotype and environment interaction, and $\varepsilon_{ijkl}$ defines the residual error.

Since all end-use quality parameters were evaluated using only one replication from each environment, the environments were considered as quasi-replications. The following model was used for analysis:

$$Y_{ijkl} = \mu + Gen_i + Rep_k + Iblock_j(Rep_k) + \varepsilon_{ijk}$$

where $\mu$ is the trait mean, $Gen_i$ is the effect of the i$^{th}$ genotype, $Rep_k$ is the effect of the k$^{th}$ replication (due to the environment), $Iblock_l$ is the effect of the l$^{th}$ iblock, and $\varepsilon_{ijk}$ defines the residual error. All variables in the two models were considered random to compute mean squares.

The entry-mean-based heritability was computed for all the traits within and across environments using the formula:

$$\text{Heritability}^2(\textit{Entry mean basis}) = \frac{\sigma^2_{gen}}{\sigma^2_{gen} + \sigma^2_{genxenv}/n.rep + \sigma^2_\varepsilon/(n.\, rep \times n.env)}$$

The BLUP values were used to calculate Pearson's correlation coefficients between all traits and to show the phenotypic plasticity of the traits across environments using boxplots in the 'ggplot2' package integrated on the R [48] (Figure S2).

### 2.4. Genotyping, Linkage Mapping, and Quantitative Trait Locus Analysis

The genomic DNA of the RILs and parents was extracted at the three-leaf stage following a modified Cetyltrimethyl Ammonium Bromide (CTAB) protocol [49]. The quantity and quality of the DNA samples were visually checked on an agarose gel by comparing them with a lambda control DNA of known concentrations. The genotyping-by-sequencing (GBS) library was constructed following the protocol by Poland et al. [50]: single-end sequencing in a NextSeq 2000 sequencer with P2-100 cycle kit (Illumina Inc., San Diego, CA, USA) and analysis using the IWGSC RefSeq v1.0 [51] as a reference for SNP calling.

In total, 23,746 SNPs were detected by GBS. Of these SNPs, 7936 were polymorphic for the RIL population. After removing the markers with missing values ≥20%, a minimum allele frequency <33.3%, heterozygote rate <10%, and false double cross-overs, 2658 polymorphic SNPs were maintained and, using JoinMap 4.0, used to construct a high-density linkage map that consisted of 35 linkage groups and covered all 21 wheat chromosomes except for chromosome 4D [52]. The QTL software IciMapping, v.4.1 [53], was used to identify traits associated with QTLs as previously described by Dhakal et al. [23] and Wang et al. [54]. The QTL analysis was performed using the phenotypic data of the traits collected from individual environments (IE), across multiple environments (MET), and their BLUPs from all environments as a combined environment (COMB) to identify traits related to QTLs. Additionally, epistatic effects, additive-by-environment interactions, and epistasis-by-environment interactions were analyzed for all the traits collected when across all-environment analyses were conducted. The BIP function in IciMapping was used to detect QTLs related to the traits within each individual environment and the COMB environment. The MET function of IciMapping was then used to determine epistatic interactions, additive-by-environment interactions, and epistasis-by-environment interactions based on the traits across all individual environments. QTLs were named following the designation '*Qtrait.tamu.chrom.Mb*' previously described by Dhakal et al. [23] and Yang et al. [26]. A consistent QTL was defined as a QTL present at the same physical position for one trait in at least two out of three (IE, MET, and COMB) analyses or two individual environments, whereas a pleiotropic QTL was defined as a QTL on the same physical position which controls two or more distinctive traits that were not highly correlated.

## 3. Results

### 3.1. Mean Performance, Variance Component Estimation, Heritability, and Correlations

Mean performances of the RIL population for all parameters across all environments were displayed by comparing their variation to the parents' variation (Table S1). For the PH, PROT, MLPT, MLTXW, KAREA, KLEN, and KPERI values, the RILs showed slightly higher mean values of the combined BLUPs than those of their parents, whereas the means for the YLD, HD, BM, BMYLD, SPM, TKW, and SHDW of the RILs were slightly lower than those of their parents. However, those differences are not statistically significant. Transgressive segregation was found for all traits, suggesting that at least some of the alleles of the genes controlling the traits in the parents are complementary (Table S2).

Variance component estimations across environments revealed significant genetic variances ($\sigma_{gen}^2$) among the RILs for all the traits ($p < 0.05$) except for MLPW, MLRS, MLTXW, BM, and BMYLD. Moreover, the combined ANOVA highlighted significant genetic-by-environment interactions (GEI) for agronomical traits, yield-related traits, and KLEN at $p < 0.01$ (Table S3 and S4).

The entry-mean-based heritability (Tables S3 and S4) varied from low (<0.30) to high (> 0.70) for various traits based on the previous classification by Hallauer et al. [55]. In general, the heritability for all the end-use quality traits ranged from low to moderate except for MLPT (0.82) and MLTXW (0.73). Moderate heritability was detected for agronomical parameters including YLD (0.32) and yield components (0.50–0.69) except for BM (0.12), BMYLD (0.17), and SPM (0.22). All grain-related traits demonstrated a high heritability (0.71- 0.85) except for KWID (0.50) (Tables S3 and S4).

A wide range of Pearson's correlation coefficients, from positive and negative, were observed for all traits. Each trait correlated with at least one of the traits, based on BLUPs across environments (Table S5 and Figure S1). Significant negative correlations were detected for YLD with HD, KAREA, KPERI, KLEN, and PROT ((−0.17)–(−0.38), $p < 0.05$–0.001). However, significant positive correlations were observed for YLD with KPS and KWID (0.16, $p < 0.05$). High and positive correlations were observed among most dough rheology parameters (0.38–0.87, $p < 0.001$) except for MLPV, which was negatively correlated with all traits ((−0.24)–(−0.39), $p < 0.01$) except MLPW. MLPT had significant positive correlations ($p < 0.001$) with MLTW, MLRS, MLTXT, and MLTXW but had negative correlations with MLPV. For yield-related traits, BM and BMYLD had significant correlations with KPS and SPM, and BMYLD was significantly correlated with TKW. While HI was significantly correlated with SHGW, KPS, and TKW (0.38–0.63, $p < 0.001$), KPS and SPM were always negatively correlated with TKW and the kernel traits. Among the kernel-related traits, a highly positive correlation was detected between DIAM and SKW (0.80, $p < 0.001$), as well as between SKW and KAREA (0.70, $p < 0.001$). KAREA had significant correlations with KLEN, KWID, and KPERI (0.50–0.89, $p < 0.001$). KPERI and KLEN showed the highest significant correlation (0.98, $p < 0.001$) in the combined analysis. Correlation matrices from individual environments with consistent trends were summarized with the results from the above BLUPs (Table S5 and Figure S1b).

Boxplots displayed the plasticity of different traits across environments (Figure S2). For grain yield (YLD), a low yield appeared in the rain-fed environments affected by drought, whereas a high yield appeared in the irrigated and rain-fed locations. In the highest yield environment (18DMS), SKW, FYLD, BM, BMYLD, HI, TKW, DIAM, MLRS, SHDW, and KPERI all had the best performance. On the contrary, HARD, PROT, ASH, MLPV, MLPW, MLTW, MLTXW, KPS, SPM, SHGW, KAREA, KLEN, and KWID demonstrated the worst performance (Figure S2).

### 3.2. Genetic Map

A total of 2658 SNPs were mapped in 35 genetic linkage groups, covering 20 of the 21 wheat chromosomes except for 4D. The genetic linkage map spanned a total length of 2333.3 cM and had an average density of 1.69 SNP/cM (Table S6). Additionally, the total physical length of the linkage groups was 7810.8 Mb with an average density of 1.36 SNP/Mb. The B genome had 1279 SNPs, while the A genome had 1081 SNPs and the D genome had 298 SNPs (Table S6).

### 3.3. QTL Identification

A total of 30 QTLs were identified, including eight for end-use quality traits, nine for agronomical traits, ten for grain yield-related traits, and eight for kernel-related traits. Through the IE, MET, and COMB analyses, these QTLs were localized on all 21 wheat chromosomes except for chromosomes 2A, 4B, 4D, 6A, 6B, 7B, and 7D (Table 1 and Figure 1). Of the 30 QTLs, 14 were consistent QTLs on 11 chromosomes, including 1A, 1B, 1D, 2B, 3A, 3B, 4A, 5A, 5D, 6D, and 7A, and 6 were pleiotropic QTLs on chromosomes 1D, 4A, 5A, 5D, and 6D (Table 1 and Figures 1, S3 and S4).

**Table 1.** Consistent and pleiotropic QTLs for end-use quality, agronomical, yield components, and kernel-related traits identified in the TAM 111/TX05A001822 population by individual and multiple environment analyses.

| QTL Name | Chr [a] | Position [b] (Mb) | Trait [c] | Trait | Environment [d] | LOD Threshold | LOD [e] | LOD (A) | LOD (A*E) | PVE [f] (%) | PVE (A) (%) | PVE (A*E) (%) | ADD [g] | SNP Alleles Increase Traits | 25LGs [h] | Peak Position [i] (cM) | QTL CI (cM) [j] | Consistent QTL [k] | Pleiotropic QTL [k] | Known Genes | Novel Genes |
|---|---|---|---|---|---|---|---|---|---|---|---|---|---|---|---|---|---|---|---|---|---|
| Qhard.tamu.1A.565 | 1A | 565.06 | HARD | Quality | 18BSP100 | 3.2 | 3.26 | - | - | 9.15 | - | - | 0.92 | TAM 111 | 2 | 62 | 60.5–62.5 | | | | y |
| Qkperi.tamu.1A.569 | 1A | 568.95 | KPERI | Kernel | 18BSP100, MET | 3.20–3.99 | 3.55–4.50 | 4.13 | 0.37 | 4.66–6.49 | 4.26 | 0.4 | 0.11–0.15 | TAM 111 | 2 | 65 | 62.5–66.5 | y | | | y |
| Qmltw.tamu.1B.616 | 1B | 615.8 | MLTW | Quality | 18DMS, MET | 3.39–4.46 | 5.22–5.56 | 2.77 | 2.79 | 7.89–8.35 | 6.56 | 1.79 | 0.54–0.97 | TAM 111 | 4 | 46 | 45.5–49.5 | y | | | |
| Qklen.tamu.1B.640 | 1B | 639.91 | KLEN | Kernel | 18BSP100, MET | 3.20–3.94 | 5.31–5.33 | 2.28 | 3.05 | 9.71–9.76 | 4.32 | 5.38 | 0.03–0.07 | TAM 111 | 4 | 63 | 62.5–63.5 | y | | | |
| Qdiam.tamu.1B.687 | 1B | 686.64 | DIAM | Quality | MET | 5.01 | 5.04 | 2.94 | 2.1 | 7.94 | 3.85 | 4.09 | 0.01 | TAM 111 | 5 | 31 | 28.5–31 | | | | y |
| Qmlpw.tamu.1D.325 | 1D | 324.51 | MLPV | Quality | 18MCG | 3.36 | 5.44 | - | - | 1.16 | - | - | −1.34 | TX05A001822 | 6 | 66 | 63.5–66.5 | | | | |
| Qmltw.tamu.1D.412 | 1D | 412.19 | MLTW | Quality | 18MCG | 3.36 | 8.59 | - | - | 20.03 | - | - | −1.72 | TX05A001822 | 6 | 75 | 74.5–75.5 | | | *Glu-D1* | |
| Qmlrs.tamu.1D.422 | 1D | 422.23 | MLRS | Quality | 17CH | 3.31 | 3.61 | - | - | 8.06 | - | - | −0.3 | TX05A001822 | 6 | 77 | 76.5–77 | | y | *Glu-D1* | |
| Qmltw.tamu.1D.422 | 1D | 422.23 | MLTW | Quality | 18DMS, MET | 3.39–4.46 | 11.58–11.79 | 1.79 | 9.79 | 16.71–18.93 | 4.31 | 12.41 | (−0.43)–(−1.50) | TX05A001822 | 6 | 77 | 76.5–77 | y | y | *Glu-D1* | |
| Qmltxt.tamu.1D.422 | 1D | 422.23 | MLTXT | Quality | COMB | 3.29 | 13.86 | - | - | 31.34 | - | - | −0.16 | TX05A001822 | 6 | 77 | 75.5–77 | | y | *Glu-D1* | |
| Qmltxw.tamu.1D.422 | 1D | 422.23 | MLTXW | Quality | 17CH, COMB | 3.29–3.31 | 3.4–4.37 | - | - | 9.17–9.51 | - | - | (−0.85)–(−1.22) | TX05A001822 | 6 | 77 | 75.5–77 | y | y | *Glu-D1* | |
| Qhd.tamu.2B.707 | 2B | 707.07 | HD | Agronomy | 18BSP100, MET | 3.2–5.12 | 5.56–7.33 | 5.72 | 1.61 | 6.84–9.30 | 4.76 | 2.07 | 0.30–0.66 | TAM 111 | 9 | 159 | 157.5–159.5 | y | | | |
| Qph.tamu.2D.16 | 2D | 15.97 | PH | Agronomy | MET | 5.05 | 5.43 | 5.1 | 0.32 | 2.97 | 2.84 | 0.14 | −0.65 | TX05A001822 | 11 | 0 | 0–2.5 | | | | |
| Qbm.tamu.3A.628 | 3A | 627.54 | BM | Yield | COMB | 3.29 | 3.93 | - | - | 10.68 | - | - | −0.44 | TX05A001822 | 14 | 14 | 13.5–14.5 | | | | |
| Qfyld.tamu.3A.654 | 3A | 653.79 | FYLD | Quality | 17CH, MET | 3.31–4.52 | 37.82–38.40 | 18.26 | 20.14 | 11.09–13.39 | 4.62 | 6.47 | (−0.79)–(−2.12) | TX05A001822 | 14 | 25 | 24.5–25.5 | y | | | |
| Qkwid.tamu.3B.567 | 3B | 566.6 | KWID | Kernel | COMB | 3.2 | 3.38 | - | - | 9.72 | - | - | 0.01 | TAM 111 | 15 | 41 | 40.5–41.5 | | | | |
| Qkwid.tamu.3B.578 | 3B | 577.61 | KWID | Kernel | DMS, MET | 3.25–4.00 | 3.81–5.00 | 4.72 | 0.29 | 8.14–9.32 | 7.34 | 0.8 | 0.01–0.02 | TAM 111 | 15 | 44 | 42.5–44.5 | y | | | y |
| Qkps.tamu.3D.24 | 3D | 23.52 | KPS | Yield | MET | 4.49 | 4.52 | 3.63 | 0.89 | 3.65 | 3.36 | 0.29 | −0.52 | TX05A001822 | 16 | 0 | 0–6.5 | | | | y |
| Qskw.tamu.3D.517 | 3D | 517.11 | SKW | Yield | MET | 5.09 | 5.21 | 4.51 | 0.7 | 3.81 | 3.75 | 0.06 | 0.29 | TAM 111 | 16 | 46 | 41.5–54.5 | | | | |
| Qklen.tamu.4A.29 | 4A | 29.27 | KLEN | Agronomy | 18BSP100, MET | 3.20–3.94 | 3.78–4.99 | 4.58 | 0.42 | 6.89–9.17 | 8.64 | 0.53 | (−0.05)–(−0.06) | TX05A001822 | 17 | 12 | 10.5–12.5 | y | y | | |
| Qph.tamu.4A.29 | 4A | 29.27 | PH | Kernel | MET | 5.05 | 6.61 | 5.72 | 0.89 | 3.51 | 3.22 | 0.29 | −0.69 | TX05A001822 | 17 | 12 | 10.5–12.5 | | y | | |
| Qhd.tamu.4A.619 | 4A | 618.93 | HD | Agronomy | 17BSP100, MET | 3.31–5.12 | 3.76–7.25 | 5.66 | 1.6 | 5.62–11.27 | 4.7 | 0.92 | (−0.30)–(−0.50) | TX05A001822 | 18 | 14 | 12.5–14.5 | y | y | *TaCWI-4A* | |
| Qtkw.tamu.4A.619 | 4A | 618.93 | TKW | Yield | 17BSP67, 17BSP100, COMB, MET | 3.20–4.52 | 3.41–8.47 | 7.49 | 0.98 | 8.84–12.31 | 8.51 | 2.87 | 0.45–0.93 | TAM 111 | 18 | 14 | 12.5–14.5 | y | y | *TaCWI-4A* | |
| Qhi.tamu.4A.621 | 4A | 621.09 | HI | Yield | COMB | 3.29 | 3.33 | - | - | 7.09 | - | - | 0 | TAM 111 | 18 | 11 | 8.5–11.5 | | | *TaCWI-4A* | |
| Qhard.tamu.4A.655 | 4A | 655.24 | HARD | Quality | 17CH | 3.31 | 3.39 | - | - | 8.49 | - | - | −1.25 | TX05A001822 | 18 | 33 | 29.5–33 | | | | y |
| Qmlpt.tamu.5A.415 | 5A | 415.44 | MLPT | Quality | MET, COMB | 3.29–4.63 | 3.37–5.76 | 5.62 | 0.13 | 6.00–8.05 | 5.87 | 0.13 | (−0.17)–(−0.21) | TX05A001822 | 21 | 60 | 59.5–60.5 | y | y | | |
| Qmltw.tamu.5A.415 | 5A | 415.44 | MLTW | Quality | MET | 4.46 | 4.82 | 4.47 | 0.36 | 11.39 | 10.67 | 0.72 | −0.69 | TX05A001822 | 21 | 60 | 59.5–60.5 | | y | | |
| Qph.tamu.5A.495 | 5A | 495.04 | PH | Agronomy | MET | 5.05 | 7.1 | 6.65 | 0.45 | 3.79 | 3.7 | 0.08 | −0.74 | TX05A001822 | 21 | 96 | 95.5–96.5 | | | | |
| Qyld.tamu.5A.532 | 5A | 531.52 | YLD | Yield | 18DMS | 3.39 | 3.88 | - | - | 6.98 | - | - | 1.72 | TAM 111 | 21 | 106 | 98.5–107.5 | | | | |
| Qph.tamu.5B.381 | 5B | 381.05 | PH | Agronomy | MET | 5.05 | 6.96 | 6.24 | 0.71 | 3.7 | 3.46 | 0.24 | −0.71 | TX05A001822 | 23 | 68 | 64.5–68.5 | | | | y |
| Qklen.tamu.5D.560 | 5D | 559.65 | KLEN | Kernel | COMB | 3.2 | 4.01 | - | - | 8.31 | - | - | −0.05 | TX05A001822 | 25 | 13 | 8.5–14 | | y | *TaCWI-5D* | |
| Qkarea.tamu.5D.560 | 5D | 560.11 | KAREA | Kernel | MET, COMB | 3.20–3.92 | 3.20–4.65 | 4.64 | 0 | 5.74–7.86 | 5.71 | 0.02 | (−0.10)–(−0.13) | TX05A001822 | 25 | 13 | 7.5–14 | y | y | *TaCWI-5D* | |
| Qkperi.tamu.5D.560 | 5D | 560.11 | KPERI | Kernel | 18BSP100, MET, COMB | 3.20–3.99 | 3.44–6.72 | 6.67 | 0.05 | 6.95–8.50 | 7.21 | 0.04 | (−0.12)–(−0.16) | TX05A001822 | 25 | 14 | 7.5–14 | y | y | *TaCWI-5D* | |
| Qskw.tamu.5D.560 | 5D | 560.11 | SKW | Yield | COMB, MET | 3.29–5.09 | 3.90–5.99 | 5.92 | 0.07 | 5.24–10.13 | 4.94 | 0.3 | (−0.28)–(−0.33) | TX05A001822 | 25 | 14 | 10.5–14 | y | y | *TaCWI-5D* | |
| Qskw.tamu.6D.26 | 6D | 26.44 | SKW | Yield | MET | 5.09 | 5.93 | 3.82 | 2.11 | 6.5 | 3.23 | 3.27 | −0.27 | TX05A001822 | 29 | 24 | 20.5–29.5 | | | | |
| Qph.tamu.6D.28 | 6D | 27.71 | PH | Agronomy | 17BSP100, MET | 3.31–5.05 | 4.42–5.55 | 3.8 | 1.75 | 10.94–2.89 | 2.08 | 0.81 | (−0.55)–(−1.23) | TX05A001822 | 29 | 25 | 18.5–32.5 | y | y | | |

**Table 1.** *Cont.*

| QTL Name | Chr [a] | Position [b] (Mb) | Trait [c] | Trait | Environment [d] | LOD Thresh-old | LOD [e] | LOD (A) | LOD (A*E) | PVE [f] (%) | PVE (A) (%) | PVE (A*E) (%) | ADD [g] | SNP Alleles Increase Traits | 25LGs [h] | Peak Position [i] (cM) | QTL CI (cM) [j] | Consistent QTL [k] | Pleiotropic QTL [k] | Known Genes | Novel Genes |
|---|---|---|---|---|---|---|---|---|---|---|---|---|---|---|---|---|---|---|---|---|---|
| Qskw.tamu.6D.28 | 6D | 27.71 | SKW | Yield | 18MCG | 3.36 | 4.29 | - | - | 7.27 | - | - | −0.71 (−0.61)–(−0.76) | TX05A001822 | 29 | 25 | 19.5–32.5 | | <u>y</u> | | |
| Qph.tamu.6D.308 | 6D | 307.97 | PH | Agronomy | MET, COMB | 3.29–5.05 | 3.81–7.44 | 7.3 | 0.14 | 3.97–11.06 | 3.96 | 0.01 | (−0.61)–(−0.76) | TX05A001822 | 29 | 59 | 48.5–60 | y | | | y |
| Qhd.tamu.7A.30 | 7A | 29.89 | HD | Agronomy | 18BSP100, MET | 3.2–5.12 | 3.41–5.66 | 4.31 | 1.35 | 4.77–5.65 | 3.53 | 1.24 | 0.26–0.52 | TAM 111 | 31 | 25 | 20.5–25.5 | y | | | |
| Qshgw.tamu.7A.577 | 7A | 576.39 | SHGW | Yield | 17BSP100 | 3.2 | 3.41 | - | - | 7.54 | - | - | −0.03 | TX05A001822 | 32 | 18 | 16.5–22.5 | | | | |

[a] Name of the chromosome based on IWGSC RefSeq v 1.0. [b] Mega base pair position based on IWGSC RefSeq v 1.0. [c] Traits: HARD—hardness index; DIAM—kernel diameter; SKW—single kernel weight; FYLD—flour yield; PROT—flour protein at 14% moisture; ASH—flour ash at 14% moisture; MLPT—midline peak time; MLPV—midline peak value; MLPW—midline peak width; MLRS—midline right slope; MLTW—midline tail width; MLTXT—midline time X time; MLTXW—midline time X width; YLD—grain yield; PH—plant height; HD—days to heading in Julian Calendar from January 1st; BM—dry biomass from hand-harvested, 0.5 m long, inner row sample; BMYLD—grain weight from BM as hand-harvested dry grain weight; HI—harvest index; KPS—kernels per spike; SPM—spikes per square meter; TKW—thousand kernel weight; SHDW—single-head dry weight; SHGW—single-head dry grain weight; KAREA—kernel area; KLEN—kernel length; KPERI—kernel perimeter; KWID—kernel width. [d] Environments: 17BSP67—2017 Bushland 67% ET; 17BSP100—2017 Bushland 100% ET; 17CH—2017 Chillicothe; 18BSP100—2018 Bushland 100% ET; 18MCG—2018 McGregor; 18DMS—2018 Dumas; 18UVL—2018 Uvalde; MET—multi-environment traits which represents data for a trait across all environments; COMB—BLUP values from combining data across all environments. [e] LOD—total logarithm of odds; LOD (A)—LOD due to additive effect; LOD (A*E)—LOD due to A*E interaction. [f] PVE—total phenotypic variance explained; PVE (A)—PVE explained by additive effect; PVE(A*E)—PVE explained by additive-by-environment interaction effect. [g] ADD—additive effects of the QTL. Positive value corresponds the favorable alleles that came from female parent TAM 111 and negative value corresponds the favorable alleles that came from male parent TX05A001822. [h] Linkage group. [i] cM—Centi Morgn distance. [j] CI—95% confidence interval of the QTL in cM. [k] Underlined "y" means that this QTL was belong to a region that associated with multiple traits. It will be consistent or pleiotropic if a single trait was talked about.

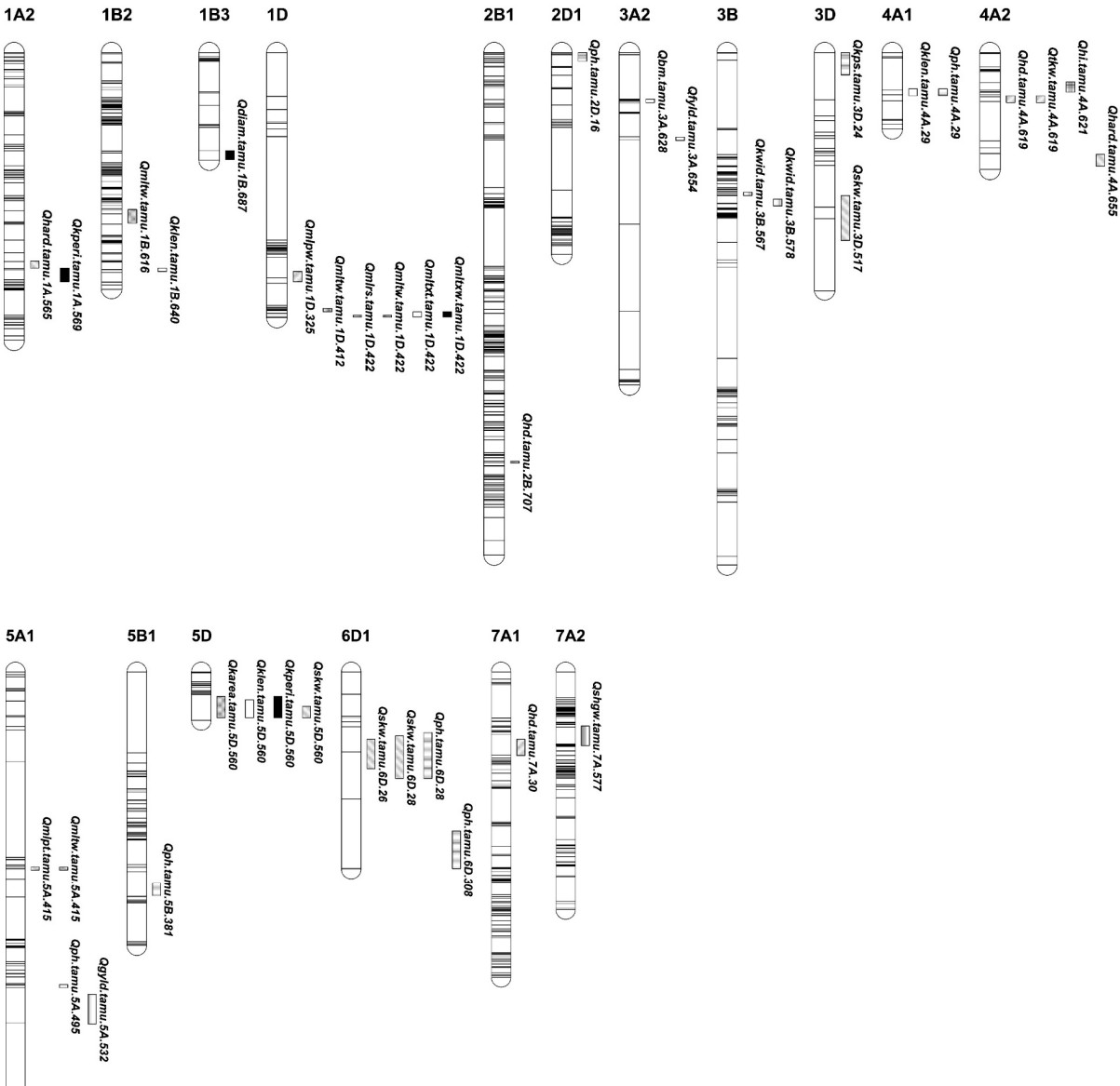

**Figure 1.** Genetic maps highlighting the positions of QTLs for end-use quality, agronomical, yield components, and kernel-related traits from the individual (IE), across multiple (MET) and combined BLUPs of the environments (COMB) in TAM 111 × TX05A001822 RIL population. Horizontal stripes inside a chromosome or linkage group represent the markers. Traits: hardness index (HARD), kernel diameter (DIAM), single kernel weight (SKW), flour yield (FYLD), flour protein at 14% moisture (PROT), flour ash at 14% moisture (ASH), midline peak time (MLPT), midline peak value (MLPV), midline peak width (MLPW), midline right slope (MLRS), midline tail width MLTW, midline time X time (MLTXT), midline time X width (MLTXW), grain yield (YLD), plant height (PH), days to heading in Julian Calendar from 1 January (HD), dry biomass from hand harvested 0.5 m long inner row sample (BM), grain weight from BM as hand harvested dry grain weight (BMYLD), harvest index (HI), kernels per spike (KPS), spikes per square meter (SPM), thousand kernel weight (TKW), single-head dry weight (SHDW), single-head dry grain weight (SHGW), kernel area (KAREA), kernel length (KLEN), kernel perimeter (KPERI), and kernel width (KWID). The designation of detected QTL was formatted as *Qtrait.tamu.chrom.Mb*. The bar length under each QTL is the flanking marker intervals in cM.

### 3.3.1. QTL for End-Use Quality

A total of eight QTLs were identified for end-use quality traits on chromosomes 1A, 1B, 1D, 3A, 4A, and 5A, with four consistent and two pleiotropic QTLs (Table 1 and Figure 1). For kernel characteristics, three minor QTLs appeared in one of the IE, MET, and COMB analyses. Two QTLs for HARD, explaining a phenotypic variation up to 9.2%, were mapped on chromosomes 1A at 565.1 Mb and 4A at 655.2 Mb with favorable alleles from TAM 111 and TX05A001822, respectively. From the MET analyses, one QTL explained a phenotypic variation (PVE) of up to 7.9% for DIAM on chromosome 1B at 686.6 Mb. A consistent QTL for FLYD was identified at 653.8 Mb on chromosome 3A from 17CH and MET. It had favorable alleles from TX05A001822 and explained a PVE of 12%.

Several QTLs were identified for dough rheological parameters. A consistent QTL for MLPT at 415.4 Mb on chromosome 5A was detected in the COMB and MET analyses. It explained 6.0–8.1% of the PVE, with favorable alleles from TX05A001822. Two consistent QTLs for MLTW were mapped at 615.8 Mb on chromosome 1B and at 422.2 Mb on 1D. The favorable allele for the 1B QTL was from TAM 111 and explained up to 8.4% of the PVE. The QTL on 1D had favorable alleles from TX05A001822 and explained up to 18.9% of the PVE. A consistent QTL linked to MLTXW at 422.2 Mb on 1D was identified with a PVE of 9.2% in 17CH, and a PVE of 9.5% in the MET analyses. Additionally, five minor rheological QTLs were identified in one of the three analyses (IE, MET, and COMB) and were found on chromosomes 1D (MLPV at 324.5 Mb, MLRS at 422.2 Mb, MLTW at 412.2 Mb, and MLTXT at 422.2 Mb) and 5A (MLTW at 415.1 Mb). All five QTLs had their favorable alleles from TX05A001822, explaining up to 31.3% of the phenotypic variations.

### 3.3.2. QTL for Agronomical Traits

Nine QTLs were associated with HD and PH, including three consistent QTLs for HD and two consistent QTLs for PH (Table 1 and Figure 1). The two consistent QTLs were mapped on chromosome 6D at 27.7 Mb and 308.0 Mb, and increased PH up to 1.2 cm with the taller alleles from TX05A001822. Four minor PH QTLs were identified in the MET analyses with taller alleles from TX05A001822. They were located at 16.0 Mb on chromosome 2D, 29.3 Mb on 4A, 495 Mb on 5A, and 381.1 Mb on 5B.

Three consistent QTLs for HD were located at 707.1 Mb on chromosome 2B, 618.9 Mb on 4A, and 29.9 Mb on 7A. The QTL on chromosome 4A had favorable alleles from TAM 111 that explained 4.7% of the phenotypic variation. The other two consistent QTLs had favorable alleles from TX05A001822 with PVE up to 4.7%.

### 3.3.3. QTLs for Yield and Component Traits

Ten QTLs associated with grain yield and its components were identified on chromosomes 3A, 3D, 4A, 5A, 5D, 6D, and 7A (Table 1 and Figure 1). The four QTLs for SKW, TKW, HI, and YLD on chromosomes 3D, 4A, and 5A had favorable alleles from TAM 111. One minor QTL for grain yield was mapped at 531.5 Mb on chromosome 5A and had its favorable alleles from TAM 111. One major consistent QTL for TKW was detected on chromosome 4A at 618.9 Mb in two single environments (17BSP100 and 18BSP100) in the MET and COMB analyses. The QTL increased TKW up to 0.93 g and explained 12.3% of the PVE. For the harvest index (HI), a minor QTL, which had PVE of up to 7.1%, was identified at 621.1 Mb on chromosome 4A. The other six QTLs had the alleles from TX05A001822 that increased SKW, KPS, and BM. A consistent QTL associated with SKW on chromosome 5D at 560.1 Mb explained up to 10.1% of the phenotypic variations. Two SKW QTLs were detected on chromosome 6D with up to 7.3% of the PVE and increased SKW by 0.71 mg. For KPS, a QTL was identified at 23.5 Mb on chromosome 3D, explaining 3.6% of the phenotypic variations. No QTL was detected for SPM. For BM, a minor QTL was identified at 657.5 Mb on chromosome 3A, explaining up to 10.7% of the phenotypic variation and increasing the BM by approximately 12.5 g m$^{-2}$.

### 3.3.4. QTLs for Kernel-Related Parameters

A total of six QTLs controlling kernel-related traits, five of which were consistent QTLs, were mapped on five chromosomes: 1A, 1B, 3B, 4A, and 5D (Table 1 and Figure 1). One consistent QTL at 560.1 Mb on chromosome 5D was associated with KAREA, KPERI, and KLEN, explaining up to 8.5% of the phenotypic variation and increasing KLEN by 0.05 mm. Another consistent QTL at 29.3 Mb on chromosome 4A increased KLEN by 0.05 mm. Both QTLs had increasing alleles from TX05A001822. The other four QTLs had the kernel-trait-increasing alleles from TAM 111. The consistent QTL for KPERI at 569 Mb on chromosome 1A increased KPERI up to 1.5 mm. The consistent QTL for KLEN at 639.9 Mb on chromosome 1B explained up to 9.8% of the phenotypic variation. In addition, two QTLs were identified for KWID on chromosome 3B, where a consistent QTL was mapped at 577.6 Mb and had a PVE of up to 9.3%, increasing the KWID by 0.02 mm.

### 3.4. Pleiotropic QTLs

Six pleiotropic QTLs that contributed to two or more traits were identified on chromosomes 1D, 4A, 5A, 5D, and 6D (Table 1). The pleiotropic QTL on chromosome 1D at 412.2–422.2 Mb was found to be responsible for four mixograph parameters, including MLRS, MLTW, MLTXT, and MLTXW. This QTL had PVE up to 31.4% and its favorable alleles came from TX05A001822. The second pleiotropic QTL was at 29.3 Mb on chromosome 4A. It was associated with KLEN and PH, with its favorable alleles coming from TX05A001822. The third pleiotropic QTL was at 618.9 Mb on chromosome 4A, and it was linked to HD and TKW. The favorable alleles for TKW, which came from TAM 111, increased TKW by 0.93 g. The fourth pleiotropic QTL was at 415. 4 Mb on chromosome 5A and was associated with MLPT and MLTW, receiving its favorable alleles from TX05A001822. The fifth pleiotropic QTL was identified at 560.1 Mb on chromosome 5D for KAREA, KLEN, KPERI, and SKW. It had favorable alleles from TX05A001822 and explained up to 10.1% of the PVE. The sixth pleiotropic QTL was at 27.7 Mb on chromosome 6D. It influenced both SKW and PH and explained up to 10.94% of the PVE, with its favorable alleles obtained from TX05A001822 (Table 1).

### 3.5. Interactions of Epistasis, Epistasis-By-Environment, and Additive-By-Environment

For all studied traits except MLTXT, BM, and KWID, a total of 376 digenic epistatic QTLs with LOD scores higher than five were detected (Table S8 and Figure S5). For end-use quality traits, 138 digenic epistatic QTLs were identified, involving one consistent and pleiotropic QTL for SKW and another pleiotropic QTL for SKW. The LOD scores for the additive-by-additive (AA) interactions for the digenic QTLs were up to 8.6, whereas the LOD scores of the additive-by-additive-by-environment (AAE) interactions were up to 5.2. The phenotypic variations explained by epistasis were not significant. For agronomical traits, 106 digenic epistatic QTLs were detected, including one consistent and pleiotropic QTL for PH, one consistent QTL for HD, and one minor QTL for PH. The LOD scores of the AA interactions were up to 10.9, whereas the LOD scores of the AAE interactions were up to 2.6. For yield and yield components, 108 digenic epistatic QTLs were identified. However, none of these QTLs were involved with the mapped QTL. Fourteen epistatic interactions increased the YLD by up to 3.4%. Regarding yield components, the LOD scores of the AA interactions were up to 8.6, while the LOD scores of the AAE interactions were up to 6.0. The phenotypic variations for the yield and yield components explained by epistasis varied from 1.7% to 6.9%. For kernel-related traits, 24 digenic epistatic QTLs were detected and involved one consistent and pleiotropic QTL for KAREA. The LOD scores of the AA interactions for the digenic QTLs were up to 8.7, while the LOD scores of the AAE interaction were not significant (<0.3). The phenotypic variation explained by the epistasis ranged from 5.2% to 9.6%.

## 4. Discussion

Previous studies highlighted that quality parameters, grain yield and its components, and morphological characteristics are all complex traits that are controlled by multiple genes, environments, and genetic-by-environment interactions [6,22–24,26,36,45,54,56,57]. The phenotype ranges of the RILs displayed transgressive segregation for all traits, indicating that the gene alleles of the traits from both parents had a positive contribution to the phenotypic variation of new genetic combinations in recombinant lines, as reported in previous studies [13,18,23,24,26,45,54,58]. Notably, most end-use quality traits were improved by the favorable alleles from TX05A001822 in the present study.

Based on their BLUP values, all traits investigated in this study displayed significant pairwise correlations in individual and across environments (Figure S1). Dough rheology parameters, measured by mixograph, showed the most significant and positive correlations to each other except for MLPV, which is consistent with previous studies [6,18–20,23,24,59–61]. Most of the yield-component parameters were highly correlated with each other, which was also consistent with the previous studies [26,36,45,62]. As expected, significant correlations were found between kernel characteristics measured by SKCS and kernel-related traits identified by grain scan in individual environments and across environments. In contrast to the findings by Tsilo et al. [18] and Dhakal et al. [23], prominent relationships between mixograph traits and kernel traits were barely observed. Furthermore, BMYLD was significantly and positively correlated with all three yield components traits; however, none were significantly correlated with the YLD. YLD was significantly correlated with KPS and KWID for the overall BLUP but not in individual environments. Moreover, YLD and PROT were negatively correlated in the overall BLUP and some individual environments that were referred to by the previous studies [13,18,63,64]. These results may assist wheat breeders in increasing the wheat protein content independently by increasing secondary yield-related traits.

The range of heritability was wide, ranging from low to high (0.12–0.87) among the traits. The estimated heritability was 0.32 for the YLD. This was in parallel with our results, which reported a comparatively low estimated heritability for the YLD [13,65]. End-use quality traits, including single-kernel characteristics, milling, and mixing properties, showed a moderate-to-high broad-sense heritability (0.32–0.82) except for two milling traits (FYLD and ASH) and one mixograph trait (MLPW). This is consistent with previous studies [6,23,24,57,61]. Additionally, the heritability of MLPT, one of the most significant mixograph parameters, was 0.82; this is similar to the previous studies [17,23,24,58,61]. Among the agronomical and yield component traits, BM, BMYLD, and SPM displayed a low heritability, while other traits were moderately heritable. This was similarly reported by Assanga et al. [45] and Yang et al. [26]. On the other hand, except for KWID (0.50), kernel-related traits demonstrated a high heritability (0.71 to 0.87).

In this study, a set of eight QTLs, including four consistent QTLs for end-use quality traits, were mapped on four chromosomes (1B, 1D, 3A, and 5A). Previous studies indicated that *Puroindoline* genes on chromosome 5D affected the genetic variation in kernel hardness [8,9]. On the contrary, the QTLs associated with HARD were only found on chromosomes 1A and 4A in the present study, which was similar to previous studies [18,23,56,66–68]. Dhakal et al. [23] showed that a QTL at 475 Mb on 1A from the RIL population derived from TAM 111 as a common parent, and Aoun et al. [68] reported that a marker-trait association (MTA) at 583 Mb on 1A in an association mapping population consisting of 672 soft white winter wheat breeding lines and cultivars. Moreover, Juliana et al. [69] reported an MTA (*Qhard.tamu.1A.565*) in a similar position (on 1A at 565 Mb) that affects protein content. On the other hand, the other minor QTLs, *Qhard.tamu.4A.655* and *Qdiam.tamu.1B.687*, have not been previously reported; therefore, they might be novel QTLs.

For FYLD, QTLs have been reported in the entire bread wheat genome except for 1A, 2B, and 3D [6,12,21,23,24,59,61,64,70]. The consistent FYLD QTL Qfyld.tamu.3A.654 was close to the previously detected FLYD QTL at 621 Mb on chromosome 3A [68].

A mixograph measures the dough rheology parameters to detect gluten strength, which corresponds to the flour quality in bread making [12]. Based on the previous studies, QTLs for mixograph dough properties have been documented on homologous chromosome group 1 carrying three glutenin loci, *Glu-A1*, *Glu-B1*, and *Glu-D1* [7,12,15–24]. These three loci encode the HMW-GS proteins with a large contribution to the dough rheology parameters [10,71,72]. In this study, QTLs for mixograph traits were detected on chromosomes 1B, 1D, and 5A. A QTL controlling both MLPT and MLTW was consistently mapped at 415 Mb on chromosome 5A and was physically close to a TKW QTL at 417 Mb on 5A in the RIL population derived from TAM 111, as reported by Assanga et al. [45]. One consistent MLTW QTL, *Qmltw.tamu.1B.615*, was mapped at 615 Mb on chromosome 1B. It shared a similar physical location with the KWID QTL at 614 Mb on 1B from TAM 112/Duster [54] and the QTL controlling TKW, YLD, and spike number (SN) at 518 Mb on 1B in a yield meta-QTL analysis study and its components [73].

A QTL cluster for dough rheology parameters was located at between 325 Mb and 422 Mb on chromosome 1D. The cluster included one and co-localized a QTL for MLTW, MLRS, MLTXT, and MLTXW at 422 Mb, similar to the previous studies [17,22–24,60,74]. Yu et al. [24] reported the genomic region at 412–414 Mb on chromosome 1D was co-localized with the *Glu-D1* loci, which have a large positive influence on mixograph parameters from CO960293-2/TAM 111. Dhakal et al. [23] studied the RIL population derived from a mutual parent (TAM 111) and addressed a QTL for mixograph traits at the interval of 412.0–418.5 Mb on chromosome 1D. Another study by Rasheed et al. [75] tried to utilize *Glu-D1* at 412 Mb via the development and validation of competitive, allele-specific PCR (KASP) assays for the gene. Moreover, an MLTW, MLRS, MLTXT, and MLTXW QTL at 422 Mb was mapped at the identical physical position to the QTL for both the test weight (TW) and YLD [36]. These studies indicated that *HMW-GS* genes affect the end-use quality by the Dx5 + Dy10 alleles of *Glu-D1* in TX05A001822, CO960293-2, and TAM 112.

In bread wheat, many previous genetic studies demonstrated the presence of a YLD QTL on chromosome 5A [13,36,45,58,76–78]. In this study, *Qyld.tamu.5A.532* was physically close to the YLD QTL at 503 Mb and 553 Mb on 5A, as reported by Yang et al. [26], and at 555 Mb on 5A, as reported by Dhakal et al. [36], which had RIL populations derived from TAM 111 that contributed the favorable alleles that increase the YLD. The QTL *Qkps.tamu.3D.24* shared a similar physical location with previously reported YLD QTL in a review paper for yield components [25]; therefore, they may be the same QTL.

The present study defined eight QTLs associated with the yield components on chromosome 3D for KPS and 4A for TKW (Table 1). The QTL *Qkps.tamu.3D.24* had not been reported previously and could be a novel QTL. The consistent and pleiotropic QTL *Qtkw.tamu.4A.619* shared a similar physical location with previously reported TKW QTL at 625Mb [79], 622 Mb [80], 600–629 Mb [73] on 4A, and a pleiotropic TKW, KPS, SPM, and GWS QTL at 622 Mb [81]. Cao et al. [25] also reported a QTL linked to TKW at the interval 610.0–616.9 Mb on 4A, co-located with the yield-associated gene *TaCWI-4A* [82,83].

A consistent SKW QTL *Qskw.tamu.5D.560* and three minor SKW QTLs were detected on chromosomes 3D and 6D. Several QTLs associated with SKW have been documented in previous studies, and they were located throughout all bread wheat chromosomes [23,24,59,82,84]. Jiang et al. [83] and Afzal et al. [82] reported that a *TaCWI-5D* gene at 557.3 Mb on 5D, very close to the consistent QTL *Qskw.tamu.5D.560* in this study. They are likely the same QTL. *Qskw.tamu.3D.517* was close to a KLEN QTL mapped in the TAM 112 x Duster population [62]. It shared similar physical locations with the TKW QTL at 516 Mb on 3D, described as stable and reliable genetic loci for yield-components [25], and the QTL at 518 Mb on 3D in a RIL population from Chuannong18 x T1208 for KLEN and DIAM [85]. The other minor QTLs, *Qskw.tamu.6D.26* and *Qskw.tamu.6D.28*, had a close physical location to the QTL for TKW, YLD, grain number, and grain filling rate in a meta-QTL analysis study [73].

Regarding kernel size-related QTLs, a consistent QTL *Qkperi.tamu.1A.569* and a minor QTL *Qkwid.tamu.3B.567* have not been previously documented, indicating that they could

be novel QTLs for KWID. The KLEN QTL at 640 Mb on 1B physically overlapped with the KLEN QTL reported in a genome-wide association study of 768 Chinese wheat cultivars [86]. Additionally, consistent QTLs at 578 Mb on 3B for KWID and at 29 Mb on 4A for KLEN and PH were close to a previously reported QTL for spike length [27,86]. Moreover, a pleiotropic and consistent QTL at 560 Mb on 5D for SKW, KLEN, KPERI, and KAREA was located at a similar position to *TaCWI-5D* at 557.3 Mb on 5D, which affected the kernel size and weight and the morphology of wheat [82,83].

Certain PH QTLs with either consistent or minor effects were identified on chromosomes 2D, 4A, 5A, 5B, and 6D in this study. A minor QTL *Qph.tamu.2D.16* overlapped with a PH QTL at 13–18 Mb that was previously reported by Ward et al. [87] and Pang et al. [86]. A PH QTL at 495 Mb on chromosome 5A was physically mapped to the same position of a previously reported QTL for PH, TKW, and HD in several studies by Li et al. [88], Cao et al. [25], and Hu et al. [27]. A consistent PH QTL at 28 Mb on chromosome 6D was mapped in the same position with a pleiotropic QTL for GN, GYLD, and TKW at 27.4 Mb that was previously reported by Liu et al. [89]. QTLs *Qph.tamu.5B.381* and *Qph.tamu.6D.308* have not been previously documented; thus, they might be novel QTLs for PH.

QTL *Qhd.tamu.2B.707* was co-localized with the previously identified gene *TaGS2-B1* at 710 Mb on 2B for nitrogen use efficiency and the shoot and root dry weight of wheat [82]. The QTL was also close to an HD QTL at 745 Mb on 2B that was reported by Li et al. [88]. *Qhd.tamu.4A.619* was a consistent and pleiotropic QTL for both HD and TKW. It was co-located with a previously reported yield-associated gene, *TaCWI-4A* [82,83]. Additionally, a consistent QTL, *Qhd.tamu.7A.30*, was mapped at a similar position with a previously reported HD QTL at 32 Mb on 7A [88].

Due to the unbalanced data for this experiment (Table S1), we could not validate many other QTLs that were only identified from one individual environment (location by year) (Table S7). Texas is part of the US High Plains, where drought is very common during the wheat-growing season. We could only expect that half of the planted yield trials might be harvested. The environments for the state-wide yield trials were very diverse in rainfall, soil types, and temperatures. All these environmental factors affected the research results.

## 5. Conclusions

A set of 179 F2:6 RILs from TAM 111/TX05A001822 were phenotyped for yield, quality, and agronomic traits in multiple field locations over two years. A set of 30 QTL regions were significantly associated with the traits, with most of those QTLs colocalized with either the known genes *Glu-D1*, *TaCWI-4A*, and *TaCWI-5D* or previously reported QTLs for the same or different traits. Three consistent QTLs for kernel traits, *Qkperi.tamu.1A.569*, *Qkwid.tamu.3B.578*, and *Qph.tamu.6D.308*, were novel. Four other QTLs, *Qdiam.tamu.1B.687*, *Qkps.tamu.3D.24*, *Qhard.tamu.4A.655*, and *Qph.tamu.5B.381*, might be novel. The SNPs linked to these QTLs can be used for the introgression of favorable alleles in breeding. Epistasis was detected, but major QTLs were not involved in significant epistasis or in interactions with environmental effects.

**Supplementary Materials:** The following supporting information can be downloaded at: https://www.mdpi.com/article/10.3390/agronomy13030689/s1, Table S1: Seven individual environments showing collected traits; Table S2: End-use quality, agronomical, yield components and seed-related traits, and average performance of the parents and RILs based on overall BLUP values across tested environments; Table S3: Restricted maximum likelihood (REML) variance component estimates, heritability, and mean performance of end-use traits across the environments; Table S4: Restricted maximum likelihood (REML) variance component estimates, heritability, and mean performance of agronomical, yield components, and seed-related traits across the environments; Table S5: Pearson correlation matrix for end-use quality, yield components, and agronomical and seed-related traits for predicted means (BLUP) derived from individual environments and across all environments; Table S6: Genetic and physical length of mapped SNPs on 35 linkage groups and 21 chromosomes; Table S7: Significant QTLs for end-use quality, agronomical, yield components and kernel-related traits detected from individual environments, across environments, and the BLUP of all environment QTL analyses;

Table S8: Additive-by-additive, additive-by-environment, and additive-by-additive-by-environment interaction effect on end-use quality, yield component, and agronomical traits; Supplemental Figure S1a: Pearson correlation coefficient matrix for end-use quality, yield-related, agronomy, and kernel-related traits for best linear unbiased predictors (BLUPs) across all environments; Supplemental Figure S1b: Pearson Correlation coefficient matrix for agronomical, end-use quality, yield component, and seed-related traits in the individual environment; Supplemental Figure S2: Boxplot analysis of end-use quality, agronomical, yield components, and seed-related traits; Supplemental Figure S3a: LOD profile and additive effects of detected QTLs for end-use quality traits in each of the environments; Supplemental Figure S3b: LOD profile and additive effects of detected QTLs for agronomical traits in each of the environments; Supplemental Figure S4: Whole genome significant LOD (A) and LOD (AbyE) profiles of quantitative trait loci for end-use quality, agronomical, yield-, and kernel-related traits based on across all the environments analyses; Supplemental Figure S5a: Epistatic interaction between QTLs for quality traits; Supplemental Figure S5b: Epistatic interaction between QTLs for agronomical traits; Supplemental Figure S5c: Epistatic interaction between QTLs for yield-component traits; Supplemental Figure S5d: Epistatic interaction between QTLs for seed-related traits.

**Author Contributions:** Conceptualization, J.C.R. and S.L.; Methodology, M.D., Z.W., M.C., J.L.V.-A., S.D., C.C., Q.X., A.M.H.I., J.C.R., H.Z. and S.L.; Software, M.D., C.C. and S.L.; Validation, M.D. and S.L.; Formal analysis, M.D., Z.W., M.C., J.L.V.-A., S.D., C.C., Q.X., A.M.H.I., J.C.R., P.S.A., G.B. and S.L.; Investigation, M.D., Z.W., M.C., J.L.V.-A., S.D., C.C., Q.X., A.M.H.I., J.C.R., A.B. and S.L.; Resources, A.M.H.I., J.C.R., A.B., P.S.A., G.B., H.Z. and S.L.; Data curation, M.D., A.B., P.S.A., G.B. and H.Z.; Writing—original draft, M.D.; Writing—review & editing, M.D., Z.W., M.C., J.L.V.-A., S.D., C.C., Q.X., A.M.H.I., J.C.R., A.B., P.S.A., G.B., H.Z. and S.L.; Visualization, M.D.; Supervision, H.Z. and S.L.; Project administration, S.L.; Funding acquisition, S.L. All authors have read and agreed to the published version of the manuscript.

**Funding:** The funding for this research was provided by the Texas Wheat Producers Board, Texas A&M AgriLife Research, the National Research Initiative Competitive Grants 2017-67007-25939, 2019-67013-29172, 2021-67013-33940, and 2022-68013-36439 from the U.S. Department of Agriculture National Institute of Food and Agriculture.

**Data Availability Statement:** The data that support the findings of this study are available from the corresponding author, S.L., upon reasonable request.

**Acknowledgments:** The authors are grateful to Shannon Baker, Jason Baker, Geraldine Opena, Bryan Simoneaux, Kele Hui, Shichen Wang, and Xiaoxiao Liu from Texas A&M AgriLife Research for the help in data requisition. The mention of trade names or commercial products in this publication is solely for the purpose of providing specific information and does not imply recommendation or endorsement by the US Department of Agriculture. USDA is an equal opportunity provider and employer. Graduate study was financially supported by a fellowship from the Ministry of National Education of the Republic of Türkiye.

**Conflicts of Interest:** The authors declare no conflict of interest.

## Abbreviations

| | |
|---|---|
| RIL | recombinant inbred line |
| QTL | quantitative trait loci |
| SNP | single nucleotide polymorphisms |
| HRWW | hard red winter wheat |
| SKCS | single kernel characterization system |
| NIR | near infra-red spectrometry |
| HMWGs | high molecular weight Glutenin Subunits |
| LMWGs | low molecular weight Glutenin Subunits |
| YLD | grain yield |
| HD | heading date |
| PH | plant height |
| HARD | kernel hardness index |

| | |
|---|---|
| DIAM | kernel diameter |
| SKW | single kernel weight |
| FYLD | flour yield |
| PROT | flour protein content |
| ASH | flour ash content |
| MLPT | midline peak time |
| MLPV | midline peak value |
| MLPW | midline peak width |
| MLRS | midline right slope |
| MLTW | midline tail width |
| MLTXT | midline time X time |
| MLTXW | midline time X width |
| BM | dry biomass from hand harvested 0.5 m long inner row sample |
| BMYLD | grain weight from BM as hand harvested dry grain weight |
| HI | harvest index; |
| KPS | kernels per spike, kernels spike$^{-1}$ |
| SPM | spikes per square meter, spikes m$^{-2}$ |
| TKW | thousand kernel weight |
| SHDW | single head dry weight |
| SHGW | single head dry grain weight |
| KAREA | kernel area |
| KLEN | kernel length |
| KPERI | kernel perimeter |
| KWID | kernel width |
| REML | restricted maximum likelihood method |
| BLUP | best linear unbiased predictors |
| CTAB | cetyltrimethyl ammonium bromide |
| ANOVA | analysis of variance |

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
