# Peer review of "QTL Analysis of Yield and End-Use Quality Traits in Texas Hard Red Winter Wheat"

_agronomy, doi:10.3390/agronomy13030689_

Round 1

Reviewer 1 Report

In this study, authors identify quite a few stable QTL or cluster linked to yield and end-use quality traits including kernel, flour, and dough mixing characteristics, yield and yield components under dryland and irrigated. The findings provided useful information for wheat breeding. overall, these outcomes are informative and interesting. However, this paper is not well written. Thus, I felt that it could be considered for publication after handle linguistics problems.

1. Consistent QTL is environmentally stable QTL?

2. The sentence is too long to understand: Three major consistent and pleiotropic QTL were detected co-localized with known genes; covering dough mixing property QTL cluster around Glu-D1 on chromosome 1D with up to explained 31.3% phenotypic variations (PVE), and kernel-related QTL physically close to the TaCWI-4A (cell wall invertase) gene influencing thousand kernel weight, heading date, and harvest index on chromosome 4A with a PVE up to 12.3 %., and colocalized with the TaCWI-5D gene for kernel traits on chromosome 5D with a PVE of 6.7%.

3. The sentence has grammatical mistakes : The current study provided provides new information useful for enhanced wheat breeding…..

Reviewer 2 Report

This work presents a regular QTL mapping of quality and yield components in wheat. The subject is not new as recognized by the authors in the discussion section, where most of their results may be contrasted with many previous studies. However, the work has merit as the authors detected new QTLs, therefore adding relevant new information to the scientific literature.

There are a few details that may be improved in the text:

1) In lines 25 and 26 (abstract), the variable "yield" appears twice in the same sentence. Please eliminate one of them.

2) In material and methods sections there is mention of 7 environments, one of them without location name (17BSP67). Not all the studied variables were measured at the same locations, so the data matrix is very unbalanced. For this reason, two additions should be made to the text:

2a) Add a table with the locations, the experiment designation, yield, quality, biomass, grain characters, etc., and in the rows please add an "x" where data were collected.

2b) Explain in the text what are the consequences of this unbalance in the dataset and how it may have influenced the results.

3) Line133 starts with "A", I think this "A" should be erased.

4) The manuscript presents a complex work and has many authors. The "contribution" section should specify better who participated in the data acquisition, in data analysis, and in data interpretation, because in the present form it is not clear who made each task.

I think with these few changes the manuscript will be ready to publish
